# Effects of Fructose and Palmitic Acid on Gene Expression in *Drosophila melanogaster* Larvae: Implications for Neurodegenerative Diseases

**DOI:** 10.3390/ijms241210279

**Published:** 2023-06-17

**Authors:** Luis Felipe Santos-Cruz, Santiago Cristobal Sigrist-Flores, Laura Castañeda-Partida, María Eugenia Heres-Pulido, Irma Elena Dueñas-García, Elías Piedra-Ibarra, Alberto Ponciano-Gómez, Rafael Jiménez-Flores, Myriam Campos-Aguilar

**Affiliations:** 1Toxicología Genética, Biología, Facultad de Estudios Superiores Iztacala, Universidad Nacional Autónoma de México, Los Barrios N° 1, Los Reyes Iztacala, Tlalnepantla 54090, Mexico; neladiaem@gmail.com (L.F.S.-C.); quirros@gmail.com (L.C.-P.); eugeniaheres@hotmail.com (M.E.H.-P.); iduenasg@gmail.com (I.E.D.-G.); 2Laboratorio de Inmunología (UMF), Facultad de Estudios Superiores Iztacala, Universidad Nacional Autónoma de México, Los Barrios N° 1, Los Reyes Iztacala, Tlalnepantla 54090, Mexico; santiago_sigrist@iztacala.unam.mx (S.C.S.-F.); alberto_ponciano@comunidad.unam.mx (A.P.-G.); jrjf@unam.mx (R.J.-F.); 3Fisiología Vegetal (UBIPRO), Facultad de Estudios Superiores Iztacala, Universidad Nacional Autónoma de México, Los Barrios N° 1, Los Reyes Iztacala, Tlalnepantla 54090, Mexico; elias_piedra_ibarra@hotmail.com

**Keywords:** *Drosophila melanogaster*, human disease model, neurodegeneration, metabolic syndrome, fructose, palmitic acid

## Abstract

One of the largest health problems worldwide is the development of chronic noncommunicable diseases due to the consumption of hypercaloric diets. Among the most common alterations are cardiovascular diseases, and a high correlation between overnutrition and neurodegenerative diseases has also been found. The urgency in the study of specific damage to tissues such as the brain and intestine led us to use *Drosophila melanogaster* to study the metabolic effects caused by the consumption of fructose and palmitic acid in specific tissues. Thus, third instar larvae (96 ± 4 h) of the wild Canton-S strain of *D. melanogaster* were used to perform transcriptomic profiling in brain and midgut tissues to test for the potential metabolic effects of a diet supplemented with fructose and palmitic acid. Our data infer that this diet can alter the biosynthesis of proteins at the mRNA level that participate in the synthesis of amino acids, as well as fundamental enzymes for the dopaminergic and GABAergic systems in the midgut and brain. These also demonstrated alterations in the tissues of flies that may help explain the development of various reported human diseases associated with the consumption of fructose and palmitic acid in humans. These studies will not only help to better understand the mechanisms by which the consumption of these alimentary products is related to the development of neuronal diseases but may also contribute to the prevention of these conditions.

## 1. Introduction

In recent decades, the chronic consumption of highly energetic ultra-processed foods has increased considerably globally [1], which are widely used in the industry due to the low costs they represent, to extend the useful life and to improve the palatability of the products [2]. Among the raw materials for ultra-processed products are fructose and palm oil (palmitic acid), which have been shown to play an important role in the progression of metabolic disorders [3,4].

Much research has shown that there are many direct and indirect associations between the consumption of ultra-processed foods with large amounts of added sugars, saturated fats, and sodium, and the development of chronic noncommunicable diseases (CNCDs) [5,6,7]. According to reports by the World Health Organization, CNCDs have caused approximately 74% of deaths worldwide, which is equivalent to approximately 41 million deaths each year and, of these, 86% (17 million people) die before the age of 70 and the majority are considered premature deaths. However, it is possible that these statistics are being underestimated, since the diseases considered as a direct or indirect consequence of the intake of hypercaloric products added with fructose and palmitic acid (Western Diet) are cancer, cardiovascular diseases, diabetes, and kidney diseases associated with diabetes, leaving out diseases such as mental disorders both mild and severe [8].

Recently, several studies have begun to explore a vast field where more associations are found between the Western diet and the early development of mental disorders such as neurodegenerative diseases, where it has been observed that an increase in the intake of diets high in carbohydrates and fats increase the risk of developing mental illnesses such as depression and anxiety [9,10,11], and close relationships have also been found with the premature onset of neurodegenerative diseases such as Parkinson’s and Alzheimer’s [12,13].

Most of the changes generated in organisms by the consumption of hypercaloric diets are reflected more quickly and directly in the small intestine as it is the organ of first exposure and where numerous works have focused [14,15,16]; however, the effects of these on other tissues and organs such as the heart, lungs, and brain have only recently begun to be explored [17]. Part of the maintenance of systemic homeostasis and the response to food intake requires communication between organs, and the deregulation of these interactions could lead to changes in metabolic pathways that lead to the development of various pathologies [18].

Therefore, association studies have recently been carried out to gain more insight into molecular markers associated with joint damage to various tissues caused by the chronic consumption of Westernized diets, such as the gut–brain axis (GBA), which shows bidirectional communication between the central nervous system and intestinal functions, showing deregulation in the production of amino acid detection receptors and amino acid transporters, which is capable of stimulating intestinal endocrine cells to release intestinal hormones, which trigger a series of physiological effects that are reflected through the nervous system in processes such as appetite regulation and energy balance [17,18,19].

Due to the need to carry out, in non-vertebrate living organisms, increasingly detailed analyses involving specific relationships between different tissues, *Drosophila melanogaster* has been widely used as a model to study differential gene expression between different tissues and treatments [20,21]. In addition, it has established itself as a very useful tool for biomedical research and the molecular mechanisms involved in human diseases, due to its short life cycle, low maintenance cost, and, especially, evolutionary conservation of biological processes, as well as its approximately 75% homology to disease-related genes in humans [22].

Numerous studies have shown that *Drosophila melanogaster* is a good model organism to study the transcriptional changes generated by dietary modifications, as demonstrated by Stobdan et al. (2019) who fed flies with a high-lipid diet and found a differential expression in the head of *D. melanogaster* with an increase in the expression of genes related to the regulation of lipid homeostasis, with respect to a normal diet [23]. Furthermore, the authors also found a strong correlation between the expression of the *takeout* gene family, related to the regulation of the circadian cycle in *D. melanogaster* [24], and hyperphagic behavior. On the other hand, Rivera et al. (2019) [25] were able to see, in the head of *D. melanogaster*, changes in gene expression induced by the addition of fat in the diet; the genes that showed a differential expression were related to important metabolic pathways in processes such as memory, metabolism, smell, and motor function, which are related to processes that predispose to Alzheimer’s disease in humans [25,26,27].

Due to all the above, the aim of this study is to be able to reveal the existence of transcriptional modifications in the brain and midgut, related to neurodegenerative diseases, resulting from the enrichment of the diet with fructose and palmitic acid in third instar larvae (96 ± 4 h) of the wild Canton-S strain of *Drosophila melanogaster*.

## 2. Results

Transcriptomic analysis showed that, by comparing gene expression in midgut cells of *Drosophila melanogaster* larvae fed a diet enriched with 1% palmitic acid and 5% fructose (overnutrition) and the gene expression in the organisms fed with the normal diet (ND), a total of 1973 genes showed a modified expression. Of these, 980 were upregulated, while 993 were downregulated (Figure 1A).

On the other hand, a total of 1265 differentially expressed genes (DEGs) were observed in brain tissue from flies fed a diet added with fructose and palmitic acid compared to brain tissue from flies fed the ND. Of all the DEG, a total of 968 genes were repressed, while 297 were found to be overexpressed (Figure 1B).

The KAAS-KEGG pathway enrichment analysis showed that there are metabolic pathways that are changed identically in both tissues by the MD. One of the pathways that was significantly modified is the synthesis of amino acids, both essential and non-essential (see Figure 2) (Appendix A).

Among the modified non-essential amino acid synthesis pathways, we found alanine, glutamine, asparagine, aspartate, and serine. All the modified synthesis pathways, both essential and non-essential amino acids, had a reduction in their transcription, both in the midgut and in the brain in organisms fed with an enriched diet with fructose and palmitic acid compared to organisms fed the ND (see Figure 2).

Other transcripts with modified expression were those encoding enzymes involved in the metabolism of alanine, aspartate, and glutamate, specifically glutamate decarboxylase (4.1.1.15), amidophosphoribosyltransferase (2.4.2.14), and glutamine synthetase (6.3.1.2). In the case of the three modified transcripts, a decrease in their expression associated with the consumption of fructose and palmitic acid was observed (Figure 3).

Tyrosine metabolism was affected in the brain and midgut with an experimental diet or enriched diet, through decreased expression of tyrosine hydroxylase (1.14.16.2) and dopachrome tautomerase (5.3.3.12) transcripts (Figure 4).

One of the biological processes with the highest number of modifications in the transcripts of its participating proteins was oxidative phosphorylation, which showed five transcripts with increased expression in overnourished larvae, compared to individuals fed with the ND. The modified products were NADH ubiquinone reductase (7.1.1.2), quinol-cytochrome-c reductase (7.1.1.8), and F1-ATPase (7.1.2.2) (Figure 5).

## 3. Discussion

In recent decades, globalization and urbanization have driven changes in the food system, favoring the substitution of the traditional diet for the Western diet, what we now know as nutritional transition [28,29]. This diet is characterized by a high consumption of processed foods rich in saturated fat (palmitic acid), trans fatty acids, sodium, as well as an excessive consumption of sugar, particularly fructose [30,31]. This feeding system has been associated with an increase in the prevalence of overweight, obesity, and other NCDs related to the intake of diets enriched with fructose and palmitic acid, such as diabetes, hypertension, coronary disease, and cancer [32,33,34]. In addition to this, it has been described that the consumption of these diets rich in palmitic acid and fructose is related to the development of mental illnesses such as anxiety, depression [35,36], as well as neurological disorders such as Parkinson’s [37] and Alzheimer’s [38], due to deregulation in the hypothalamus–pituitary–adrenal (HPA) axis [39,40,41]. However, the impact of this type of diet on the synthesis of amino acids as neurotransmitter precursors and metabolic regulators has been poorly documented. In our work, we found that the administration of a diet enriched with 1% palmitic acid and 5% fructose modified the transcriptional expression that most likely alters the synthesis of essential amino acids (isoleucine, leucine, valine, and threonine) in the brain and midgut of *Drosophila melanogaster* (Figure 2).

These amino acids are related to energy homeostasis since their decrease causes a reduction in fat mass and body weight [42]. In our model, the reduction in these amino acids may be part of a balancing system, since these responses tend to drive an increase in energy expenditure and lipolysis, something that would be required in a phenomenon where there is an excess of energy. It is not surprising that these modifications occur in association with the intake of the MD in the midgut of flies, since there are many amino acid sensors in this tissue [17]. These sensors modify their expression in intestinal endocrine cells, which triggers alterations in energy homeostasis through the nervous system [17].

In the specific case of the decrease in isoleucine and leucine, it has been shown that the decrease in their synthesis can occur in patients with neurobiological disorders such as depression, and they have even been proposed as biomarkers of these alterations [43]. Furthermore, a correlation has been reported between overnutrition and the development of depression and anxiety [44,45]. Thus, the decrease observed in our experimental organisms could explain the previously described relationship between fructose and palmitic acid consumption and the development of neuronal alterations.

Another amino acid that showed a decrease in the expression of its possible synthesis pathways in our model, both in the midgut and in the brain, was threonine. This essential amino acid is an important nutritional modulator that influences the intestinal immune system through complex signaling networks such as MAPK (mitogen-activated protein kinase) and TOR (target of rapamycin) [46]. Threonine deficiency has been linked to the development of intestinal inflammation, as it leads to a decrease in the number of intestinal goblet cells and mucin content, which impairs the function of the mucin barrier, normally responsible for preventing damage caused by digestive enzymes and microorganisms [46].

Therefore, the decrease observed in the transcriptional expression and possible synthesis pathway of this amino acid in the midgut of flies’ diet added with fructose and palmitic acid could suggest that the amounts of these elements could be related to the development of digestive disorders, such as irritable bowel syndrome and recurrent infections, phenomena present in individuals with overnutrition [47,48].

Regarding the reduction in threonine synthesis in the brain of our experimental model, previous work has shown that threonine levels are closely linked to glycine levels in nervous tissue [49]. The decrease in glycine has a direct effect on the balance of neurotransmitters, which causes alterations in the brain stem in rats [49]. This has been associated with diseases such as the impaired regulation of attention, emotions, and cognitive control, which are common in patients with overeating [50]. Therefore, the decrease in threonine induced by the MD could be related to the decrease in glycine and the mechanisms that lead to the development of these alterations in patients with overfeeding [50].

In addition to possible alterations in essential amino acid synthesis, the diet added with fructose and palmitic acid also produced alterations in non-essential amino acid synthesis pathways such as alanine, glutamine, asparagine, aspartate, glutamine, and serine (Figure 2). A decrease in serum glycine levels has been reported in patients with insulin resistance and risk factors for the development of type 2 diabetes as well as non-alcoholic fatty liver, where it has been suggested that one of the causes is a decrease in biosynthesis of this amino acid [30,51], a proposal that coincides with our results and strengthens the evidence of glycine having a role in basal metabolism [52].

It is important to point out that the role of glycine in the balance of neuronal excitatory and inhibitory signals is of great importance in cognitive function and in the repair of brain damage [53]. These functions could be altered by the decrease in the synthesis of this amino acid associated with fructose and palmitic acid intake that our results show.

In our model, we identified other neuronal regulation systems such as the GABA (gamma-aminobutyric acid) system. The synthesis of GABA is carried out in the central nervous system through the decarboxylation of glutamic acid by the action of glutamic acid decarboxylase (4.1.1.15), a pathway that requires the activity of glutamine synthase (6.3.1.2) and glutamic acid decarboxylase (4.1.1.15), which present a decreased expression in the overnourished organisms (Figure 3).

Brain function is specifically related to glutamate synthase activity [54,55], so it is not surprising that decreased expression in astrocytes leads to neonatal death in mice [54,55]. In the case of humans, deficits in the activity of this enzyme contribute to neuropsychiatric disorders such as temporal lobe epilepsy, Alzheimer’s disease, and schizophrenia [54,55], which have been related to a high consumption of carbohydrates and lipids [56]. Our data indicate that fructose and palmitic acid intake causes a decrease in glutamate synthase expression, which could in turn contribute to the development of neurofunctional abnormalities. Similar decreases have been detected in humans with neuropsychiatric disorders such as Alzheimer’s and schizophrenia.

Modifications in the expression of the products of the GABAergic system in the midgut seem to be a difficult phenomenon to correlate, since this system is commonly associated with nervous tissue. Nonetheless, the literature has confirmed that the GABAergic system functions as a mediator of the enteric nervous system, controlling intestinal function [57,58]. However, the specific regulatory mechanisms are still under investigation.

GABAergic signals in the intestine have been shown to modulate both motor activity and secretion of the digestive system [57,59,60]. Moreover, it has been suggested that GABAergic cells could have an important role in neuroimmune interaction, modulating the activity of immune cells associated with different inflammatory, systemic, and enteric conditions, such as inflammatory bowel disease [48,57]. Likewise, an association has been observed between the gastrointestinal tract and the appearance of alterations in the nervous system, such as Parkinson’s disease [61]. Therefore, the modification of this system, both in nervous and intestinal tissue, in our model agrees with published evidence about the effects of the consumption of fructose and palmitic acid and contributes to showing possible mechanisms that explain the complexity of the appearance of this type of alteration in humans with overfeeding [48,57,59].

Another of the best-known systems in the regulation of the nervous system that we identified with modifications in its expression in the brain was the synthesis of products of the dopaminergic pathway, specifically, a decrease in the transcriptional expression of tyrosine hydroxylase (1.14.16.2) (Figure 4). Tyrosine hydroxylase is essential for the biosynthesis of catecholamine, dopamine, norepinephrine, and epinephrine [62,63]. It has been reported that the congenital deficiency of this enzyme leads to the development of a neurometabolic disorder that conditions the development of childhood Parkinson’s [62]. However, the inheritance of this autosomal recessive disorder is not the only condition that leads to decreased tyrosine hydroxylase activity, since it has been documented that Western diets rich in lipids and sugar can alter the synthesis of enzymes such as tyrosine hydroxylase, favoring muscular dystonia, neuroinflammation, and oxidative stress in the nervous system [64,65].

Although, at first sight, it may seem that the alterations in the dopaminergic system are not relevant in the digestive tissue, the literature has described the existence of the gut–brain–dopamine (GBD) axis. This axis not only represents the caloric intake regulation system [66] but has also been associated with disorders related to neurotransmitter synthesis, such as metabolic syndrome and inflammatory bowel disease [67,68], as well as other disorders not so commonly associated with the consumption of high-fat diets, such as Parkinson’s disease [37]. Hence, the common alterations in the mechanisms associated with tyrosine metabolism, found both in the digestive tract and in the brain of flies fed with the MD, represent possible mechanisms that relate to the excessive consumption of these alimentary products with diseases related to this process.

Finally, we found that the administration of the diet enriched with palmitic acid and fructose modified the transcriptional expression of several enzymatic components of the electron transport chain, suggesting an alteration in the redox state and a possible oxidative stress in the nervous tissue and gastrointestinal tract (Figure 5). This phenomenon has been widely documented because of hypercaloric diets [69,70], and has been related to the development of changes in the nervous system [71,72]. It has been proved that the imbalance in the redox state in the nervous system could not only be an effect of obesity but could also participate in the development and permanence of this alteration [73]. Our results indicate that our diet in *D. melanogaster* larvae with palmitic and fructose produces alterations in the electron transport chain and the redox state of the nervous system, which could participate in the appearance of obesity, one of the alterations most associated with overnutrition, which in turn affects both the nervous and digestive systems.

In summary, our data infer that the consumption of diets rich in palmitic acid and fructose can alter the biosynthesis of proteins at the mRNA level that participate in the synthesis of essential and non-essential amino acids, as well as fundamental enzymes for the dopaminergic and GABAergic systems in the digestive system and brain. These demonstrated alterations in the tissues of flies fed with fructose and palmitic acid may help explain the development of various reported human diseases associated with overfeeding, such as depression, anxiety, irritable bowel syndrome, recurrent digestive tract infections, impaired control of emotions and cognition, Alzheimer’s disease, and schizophrenia.

Therefore, we consider that further studies are needed to confirm that these modifications in humans are associated with fructose and palmitic acid consumption. These studies will not only help to better understand the mechanisms by which the intake of these products is related to the development of neuronal diseases but may also contribute to the prevention of these conditions.

## 4. Materials and Methods

### 4.1. Culture of the Wild Canton-S Strain of Drosophila melanogaster

For this study, the midgut and brain of third instar larvae (96 ± 4 h) of the wild Canton-S strain of *Drosophila melanogaster* were used, which were fed two different diets for 24 h: a normal diet (ND), which was considered a control diet, and a mix diet (MD) added with both 5% fructose and 1% palmitic acid, considered as the test diet, in order to simulate a Western human overnutrition diet.

It is important to mention that, for the flies fed with a MD, three parental generations were maintained in the ND to erase the possible epigenetic footprint that could interfere with the results [74], and all larvae were fed with the MD after 72 ± 4 h of development.

### 4.2. Culture Media

The normal diet (ND) is a modification of the medium established by Hemphill et al. (2018) [75]. The composition for a 100 mL preparation was: 2.6% yellow cornmeal (Genesee Scientific 62-100^®^, Morrisville, NC, USA), 4% inactive dry yeast (Genesee Scientific 62-106^®^), 2% agar (Genesee Scientific 66-103^®^), 3% sucrose, 1.5% methyl paraben (Apex, Energetics, Irvine, CA, USA), and 0.3% propionic acid (Reasol, Mexico City, Mexico). For the Mix Diet (MD), an additional 2% agar was added to the 2% already existing in the ND, 1% palmitic acid, and 5% fructose.

Based on Hemphill et al. (2018) [75], who studied diets high in coconut oil and sucrose, and Wang et al. (2013) [76] and Shi et al. (2021) [77], who investigated palmitic acid at 2% and 3%, respectively, the percentages of palmitic acid (1%) and fructose (5%) for the MD were calculated in three previous independent experiments with treatments, from eclosion until pupation, with the ND, palmitic acid (3.2%), fructose (5%), five replicates/tubes per treatment, and one couple of flies per replicate/tube. The percentage of palmitic acid was reduced to 1%, avoiding altering the percentage of emergence vs. control (ND) and evaluating the consistency of the medium [75,76,77].

### 4.3. Removal of Midgut and Brain Tissues

Third instar larvae of the ND and MD had their midgut or brain removed in cold PBS (4 °C) and placed in trizol at −70 °C, taking no more than 30 min throughout the process, to avoid changes that could interfere with the results.

Once the tissues were isolated, the number of organs necessary to obtain the optimal total RNA concentration for transcriptome profiling by RNAseq was standardized. Thus, extractions with 100 and 200 organs were tested for purity, integrity, and RNA concentration, establishing that the optimal number of isolated organs was 100, for both the midgut and the brain.

### 4.4. RNA Extraction

For RNA extraction, the isolated tissues were processed using the RNeasy Kit (QIAGEN) following the manufacturer’s instructions. To guarantee the sequencing work, the messenger RNA (mRNA) was purified with the RNeasy Pure mRNA Bead Kit (QIAGEN, Hilden, Germany). The mRNA quantification was performed using a NanoDrop 200. All the samples obtained met the following conditions: concentration ≥ 5.0 ng/µL, volume ≥ 20 μL, and purity in the OD260/280 ratio = 2. RNA extraction was performed in duplicate for each diet and for each of the tissues, giving a total of eight samples.

### 4.5. Bioinformatic Analysis

Transcriptome profiling of the eight samples was performed using RNAseq, configured on the Illumina Nova Seq 6000 platform, carried out by Novogene Corporation Inc. (Sacramento, CA, USA). The techniques and parameters described in this subsection for quality control, gene expression analysis, and statistical analysis are those implemented and provided by Novogene Corporation Inc. (www.novogene.com, accessed on 16 May 2020).

Once the sequencing was performed, the raw data in FASTQ format were analyzed using fastp, where the reads were cleaned along with the calculation of Q20 (the proportion of bases with a phred base quality score of Q20; the proportion of bases read with less than a 1% error rate), Q30 (the proportion of bases with a phred base quality score of Q30; the proportion of bases read with less than a 0.1% error rate), and the GC content (the ratio of G and C base numbers to total bases); all subsequent analyses were based on these calculations and the clean sequences [78].

Sequence mapping was performed by means of an alignment based on a *Drosophila melanogaster* reference id = 204923 genome obtained directly from the genome website browser (NCBI/UCSC/Ensembl) (https://www.ncbi.nlm.nih.gov/genome/47?genome_assembly_id=204923, accessed on 20 May 2020) using HISAT2 software version 2.2.1 [79].

StringTie was used to assemble the set of transcript isoforms from each .bam file obtained in the mapping. The GffCompare program can compare StringTie assemblies with reference annotation files and help separate new genes from known ones; subsequently, featureCounts was used to count the mapped read numbers of each gene, including known and new genes [80]. The RPKM (reads per kilobase of exon pattern per million mapped reads) of each gene was then calculated based on the length of the gene and the count of reads assigned to that gene. RPKM considers the effect of sequencing depth and gene length for read counts and is currently the most widely used method for estimating gene expression levels.

Differential expression analysis between two groups was performed using the R DESeq2 package (http://www.bioconductor.org/packages/release/bioc/html/DESeq2.html, accessed on 21 May 2020). The resulting *p* values were adjusted using the Benjamini and Hochberg approach to control for false discovery rate (FDR). Genes with an adjusted *p* value < 0.05 found by DESeq2 were assigned as differentially expressed.

Enrichment analysis was performed using the clusterProfiler R package [81] to test for statistical enrichment of differentially expressed genes in KEGG pathways (http://www.genome.jp/kegg/, accessed on 25 May 2020). It is important to note that the metabolic pathways and genes reported in this work correspond to functional orthologous genes in humans.

## Figures and Tables

**Figure 1 ijms-24-10279-f001:**
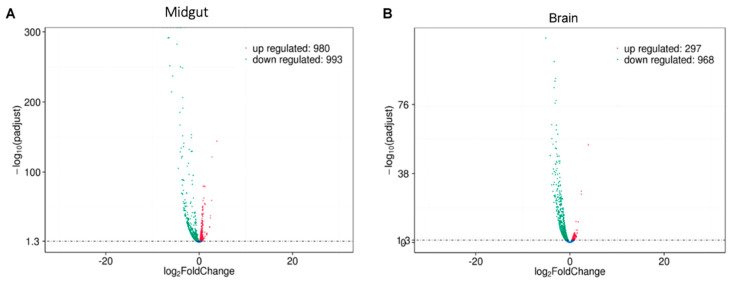
Volcano plot of the differential gene expression analysis between Normal Diet (ND) and Mix Diet (MD). (**A**) Midgut tissue showed 980 upregulated genes (red dots) and 993 downregulated genes (green dots). (**B**) The brain tissue showed 297 upregulated genes (red dots) and 968 downregulated genes (green dots). The blue dots represent those genes that did not show statistically significant differential expression.

**Figure 2 ijms-24-10279-f002:**
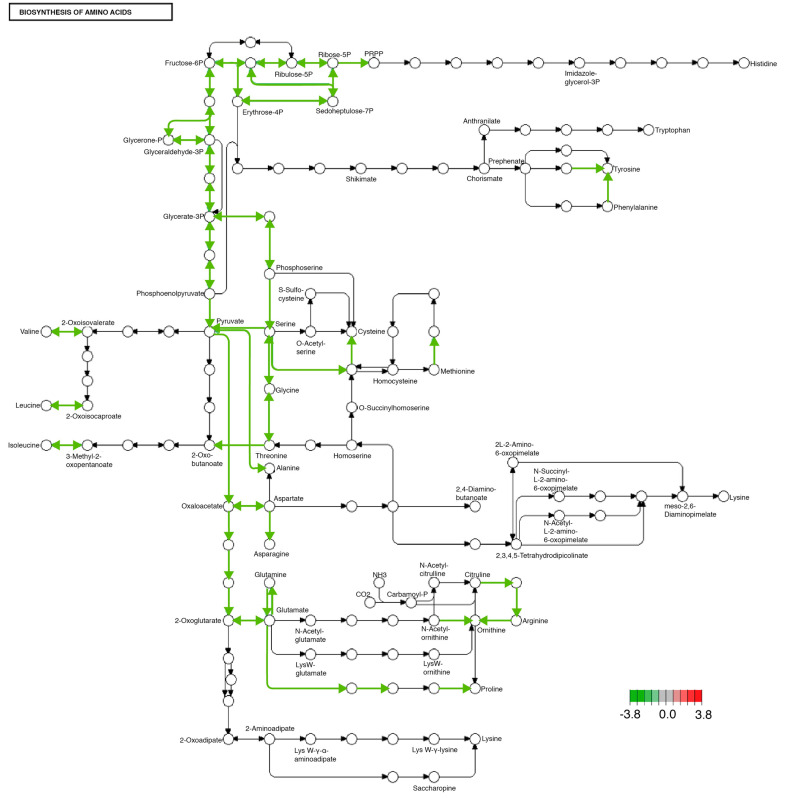
Significantly differentially expressed genes identified by KEGG in the biosynthesis of amino acids pathway. Green lines indicate significantly downregulated genes involved in this pathway. This pathway was identically modified in both midgut and brain tissues.

**Figure 3 ijms-24-10279-f003:**
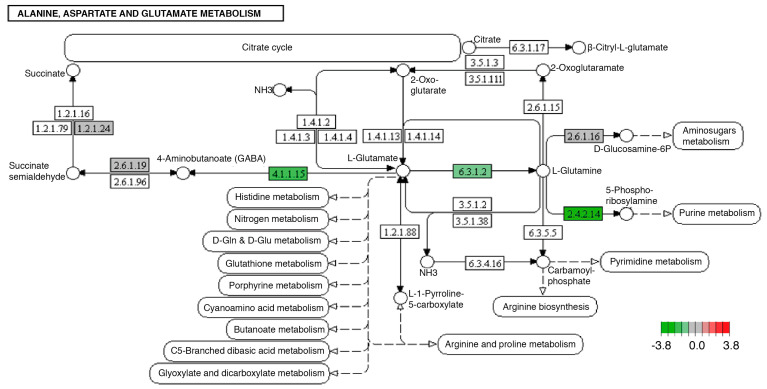
Significantly differentially expressed genes identified by KEGG in the alanine, aspartate, and glutamate metabolism pathways. Green boxes indicate significantly downregulated genes. This pathway was modified identically in both midgut and brain tissues.

**Figure 4 ijms-24-10279-f004:**
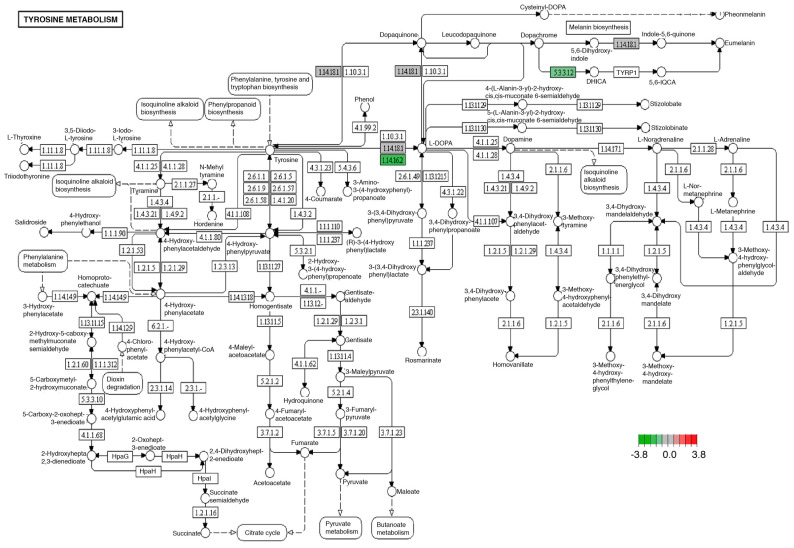
Significantly differentially expressed genes identified by KEGG in the tyrosine metabolism pathway. Green boxes indicate significantly downregulated genes. This pathway was identically modified in both midgut and brain tissues.

**Figure 5 ijms-24-10279-f005:**
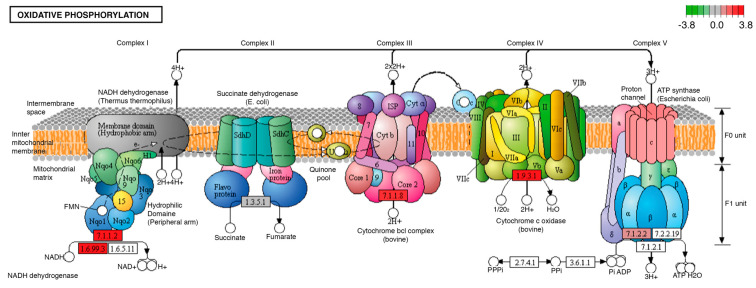
Significantly differentially expressed genes identified by KEGG in the oxidative phosphorylation pathway. Red boxes indicate significantly upregulated genes. This pathway was identically modified in both midgut and brain tissues. The letters I, II, II, III and IV correspond to the number of the enzyme complexes of the mitochondrial respiratory chain: Complex I or NADH dehydrogenase; Complex II or Succinate dehydrogenase; Complex III or Cytochrome bcl; Complex IV or Cytochrome C oxidase and Complex V or ATP synthase, as shown in the picture. The letters a, b, c, as well as α, β, γ and δ, refer to the subunits that are part of complex V or ATP synthase of the mitochondrial respiratory chain.

## Data Availability

Data are available on Figshare platform, under the link https://figshare.com/account/home#/projects/167981 (accessed on 31 May 2023).

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
