# Peer review of "Effects of Fructose and Palmitic Acid on Gene Expression in Drosophila melanogaster Larvae: Implications for Neurodegenerative Diseases"

_ijms, 2023, doi:10.3390/ijms241210279_

Round 1

Reviewer 1 Report

In this manuscript, the authors investigate the potential link between a diet rich in fructose and palmitic acid and transcriptional modifications that may lead to neurodegenerative diseases. Through this study, they aim to shed light on the possible underlying mechanisms, and ultimately contribute to the development of effective prevention and treatment strategies.

Overall, the manuscript is well written and features clear experimental design, however, following suggestions could further improve the manuscript.

-       In Figure 2, perhaps make the colored arrows stands out more for enhanced readability.

-       For section 322-330 and 364-366, include corresponding data.

Overall, the language is okay, but some errors has been noticed in abstract and the first two paragraph of introduction.

Author Response

-       In Figure 2, perhaps make the colored arrows stands out more for enhanced readability.

Response: Thank you for the insightful observation from Reviewer 1. The thickness of the lines and arrows in Figure 2 has been adjusted, while the reference color has been maintained. This modification has also been incorporated into the main body of the text.

-       For section 322-330 and 364-366, include corresponding data.

Response: In response to Reviewer's accurate comment on the addition of the Drosophila melanogaster reference genome used, as initially mentioned in lines 364-366, we have incorporated both the ID and web address. These updates are now located in lines 372-374.

Considering the thoughtful comment from the reviewer concerning the text from lines 322-330, we have introduced new bibliographic citations. These adjustments can now be found on lines 330-337.

Reviewer 2 Report

High-sugar and high-fat diets are one of the leading causes of obesity, prevalent in Western societies. These types of diets have been associated with the development of several other disorders including cardiovascular diseases, cancer, and neurodegenerative disorders, but the link between diet and these conditions is still unclear. In this work, Santos-Cruz et al. use Drosophila melanogaster to dive into the mechanisms behind this link by using transcriptomic profiling. The results are interesting, and I believe the manuscript would benefit by addressing some of the points below.

Some positive comments first: 

- This is a nicely crafted work with the results being of interest to a broad audience. 

- There is a lack of descriptions across organs, as many studies have focused on these diets in the context of Nonalcoholic fatty liver disease looking at hepatic tissue. Thus, the authors’ painstaking task of using brains and midguts for their analysis is greatly appreciated. 

- The discussion is thorough. The authors have made an excellent job expanding the main results from their RNA-sequencing analysis to what is known in the context of neurodegenerative diseases.  

Some points to consider: 

- I am not sure how the optimal concentration of fructose and palmitic acid was determined. There is an indication of the use of LC50 to establish this in the Methods section, but no data or references are provided. Also, a reference for the use of palmitic acid instead of coconut oil (with a higher concentration of lauric acid) as in other studies, might be useful.  

- Does the diet the authors used in the study influence any physical landmarks such as size, weight, etc? Are the changes in transcription preceding any physiological changes such as circulating sugar levels?  

- Eight samples were processed, each extraction in duplicate, but the number of brains/midgut was omitted. Also, is there any control by sex? This is especially important considering the sex-specific differences observed under different diets on adult flies (Stobdan et al., PLoS ONE, 14(3): e0213474).

- Across the text, the authors made several asseverations that are not necessarily true. For instance, in lines 181-182, the authors stated, “we found that the administration of a diet enriched with 1% palmitic acid and 5% 181 fructose modified the synthesis of essential amino acids (isoleucine, leucine, valine, and threonine) in the brain and midgut of Drosophila melanogaster”. This entire work is based on transcriptomics which may or may not translate to changes in the proteins and may or may not influence the concentration of the end products. No measurements of isoleucine, leucine, valine and/or threonine were made to support this claim. Please, make sure to refine the text to take this into account as it can be totally misleading. 

- I am unsure if the authors plan to make these datasets available. Also, I am unsure what the privacy or ethical restrictions indicated in the Data Availability Statement could be in a study that does not deal with human samples or similar ones.   

- It might be because of the formatting, but some “D. melanogaster” are not in italics. Please check.  

Author Response

- I am not sure how the optimal concentration of fructose and palmitic acid was determined. There is an indication of the use of LC50 to establish this in the Methods section, but no data or references are provided. Also, a reference for the use of palmitic acid instead of coconut oil (with a higher concentration of lauric acid) as in other studies, might be useful.  

Response: Thanks to the reviewer's astute observation, we have recognized an error in Section 4.2 (Culture Media) regarding the conduct of an LC50 test, which, in fact, we did not carry out. Our methodology was informed by the work of Hemphill et al., 2018, which explored diets high in coconut oil and sucrose, and the studies of Wang et al., 2013, and Shi et al., 2021, which both investigated concentrations of palmitic acid at 2% and 3%, respectively. Based on these studies, we calculated the percentages of palmitic acid (1%) and fructose (5%) for the MD through three separate experiments. Each experiment consisted of three treatments (DN, palmitic acid, fructose), with five replicates/tubes per treatment, and one pair of flies per replicate/tube. We adjusted the percentages of palmitic acid and fructose, selecting those that did not impact the emergence percentage compared to the control, and evaluated the consistency of the medium. Following the reviewer's helpful suggestion, we have updated this information, which is now accurately reflected in lines 330-337.

- Does the diet the authors used in the study influence any physical landmarks such as size, weight, etc? Are the changes in transcription preceding any physiological changes such as circulating sugar levels?  

Response: Due to the reviewer's wise observation, we realized that it was necessary to specify to him that a Drosophila climbing test was performed, in which we found an important impact of MD on larval weight and emergence percentage with a notable alteration in sex ratio-primarily affecting males. The analyses of these tests suffered a delay, due to the work stoppage we had during the COVID-19 pandemic, however, these data are part of an article in progress.    

- Eight samples were processed, each extraction in duplicate, but the number of brains/midgut was omitted. Also, is there any control by sex? This is especially important considering the sex-specific differences observed under different diets on adult flies (Stobdan et al., PLoS ONE, 14(3): e0213474).

Response: Considering Reviewer 2's comments, we have provided more details on the standardization process and the quantity of both brain and midgut organs used in Section "4.3 Removal of Midgut and Brain Tissue", as noted in lines 345-346.

Regarding sex differentiation in Drosophila melanogaster third instar larvae, it's crucial to acknowledge the complexities of studying sex differences at this developmental stage. Specific methodologies and markers are required to discern the sex of larvae, often involving staining techniques or genetic analysis. However, such methods could potentially disrupt the findings of a transcriptomic analysis, as these procedures could induce stress in the subjects and consequently alter the outcomes. This stands in contrast to sex determination in adult flies, which is largely a visual process and thus, more straightforward, and quicker.

- Across the text, the authors made several asseverations that are not necessarily true. For instance, in lines 181-182, the authors stated, “we found that the administration of a diet enriched with 1% palmitic acid and 5% 181 fructose modified the synthesis of essential amino acids (isoleucine, leucine, valine, and threonine) in the brain and midgut of Drosophila melanogaster”. This entire work is based on transcriptomics which may or may not translate to changes in the proteins and may or may not influence the concentration of the end products. No measurements of isoleucine, leucine, valine and/or threonine were made to support this claim. Please, make sure to refine the text to take this into account as it can be totally misleading. 

Response: We appreciate Reviewer 2's insightful observation, and as a result, we have chosen to adjust parts of our text to explicitly state that our findings are predicated on gene expression data, rather than measurements of protein concentrations. These specific amendments can be located on lines 182-183, 201, 210, 223, and 289.

- I am unsure if the authors plan to make these datasets available. Also, I am unsure what the privacy or ethical restrictions indicated in the Data Availability Statement could be in a study that does not deal with human samples or similar ones.   

Response: We have uploaded the raw sequencing data to the Figshare platform, under the link https://figshare.com/account/home#/projects/167981. This is shown in the body of the text in section 408-409.

- It might be because of the formatting, but some “D. melanogaster” are not in italics. Please check.  

Response: In response to Reviewer 2's comment, we have undertaken a thorough review of our manuscript and have corrected instances where "D. melanogaster" was not appropriately italicized.

Reviewer 3 Report

In the manuscript titled "Effects of fructose and palmitic acid on gene expression in Drosophila melanogaster larvae: Implications for neurodegenerative diseases," the authors conducted an investigation into the metabolic consequences of overfeeding in specific tissues, namely the brain and midgut, utilizing Drosophila melanogaster as a model organism. The primary objective of the study was to explore the potential correlation between overfeeding and chronic non-communicable diseases, including cardiovascular and neurodegenerative conditions. By employing transcriptomic profiling, the study revealed that overfeeding can induce modifications in mRNA-level of proteins involved in amino acid biosynthesis, as well as key enzymes associated with the dopaminergic and GABAergic systems in the midgut and brain. These alterations provide insights into the pathogenesis of various human diseases linked to overfeeding and offer potential avenues for preventive interventions. Although the study is intriguing, several significant concerns require attention.

Major comments:

1. The study focused solely on transcriptomic analysis to identify differentially regulated genes in D. melanogaster. However, it would be beneficial for the authors to specify the corresponding human homologs of these genes and investigate any reported implications in metabolic disorders resulting from excessive food intake.

2. It is essential for the authors to document any observed physiological changes in the flies fed the high calory diet, such as alterations in weight or body size.

3. The authors frequently employ the term "overeating" throughout the paper. To validate the actual consumption of significantly larger amounts of food in the test diet compared to the normal diet, it is advisable to conduct consumption assays.

Minor comments:

1. In line 27 and 293, the correct formatting for D. melanogaster is to write it in italics.

2. In the abstract, the term "overeating" appears to be overly generalized since the authors are specifically examining the effects of excessive fructose and palmitic acid consumption, rather than the overconsumption of any other diet. Therefore, a more appropriate word should be used.

3. In line 99, please include the references Ganguly, Dey et al., 2021 and May et al., 2019.

4. It would be advantageous to present the upregulated and downregulated genes in a table or as a heatmap. This would enhance comprehension and facilitate data interpretation.

The manuscript is well written.

Author Response

Major comments:

  1. The study focused solely on transcriptomic analysis to identify differentially regulated genes in D. melanogaster. However, it would be beneficial for the authors to specify the corresponding human homologs of these genes and investigate any reported implications in metabolic disorders resulting from excessive food intake.

Response: In response to the reviewer's insightful observation, for which we are grateful, we have adjusted the wording in our manuscript. We now specify that the transcripts under discussion correspond to homologous genes in humans. Therefore, all discussions are grounded in disorders studied in humans, as identified through the KAAS-KEGG pathway enrichment analysis, as seen in lines 391-392.

  1. It is essential for the authors to document any observed physiological changes in the flies fed the high calory diet, such as alterations in weight or body size.

Response: We appreciate the reviewer's comment, and in this regard, we would like to point out that the MD reduces the larva weight and the percentage of emergence, altered the sex ratio, mostly affecting males, affected behavior as we observed through the Drosophila climbing test, but these data are part of an article in progress. On the other hand, we perform a clinical analysis of glucose, triglyceride, uric acid, and total lipid levels in homogenates from 3rd instar larvae exposed to these diets was being standardized, however, the COVID 19 pandemic interrupted these studies, which we are now taking up again and preparing for a future article.

  1. The authors frequently employ the term "overeating" throughout the paper. To validate the actual consumption of significantly larger amounts of food in the test diet compared to the normal diet, it is advisable to conduct consumption assays.

Response: Indeed, we frequently use the term 'overeating' throughout our work. Regrettably, it isn't possible for us to implement the suggested trials to confirm an increase in consumption of the provided diet. However, to address this comment, we propose changing 'overeating' and 'overfeeding' to 'overnutrition'. This is because we know the added percentage was 1% palmitic acid and 5% fructose compared to the control group, a descriptor that more accurately reflects the diet used in this study.

We will restructure the paper where the terms 'overeating' or 'overfeeding' are used, with one exception. We will maintain the term 'overeating' in the citation of the article by Kung, P.-H.; Soriano-Mas, C.; Steward, T. titled 'The Influence of the Subcortex and Brain Stem on Overeating: How Advances in Functional Neuroimaging Can Be Applied to Expand Neurobiological Models to beyond the Cortex. Rev Endocr Metab Disord 2022, 23, 719-731, doi:10.1007/s11154-022-09720-1', as the original author uses the term 'overeating'.

Minor comments:

  1. In line 27 and 293, the correct formatting forD. melanogasteris to write it in italics.

Response: In response to the reviewer's comment, we have thoroughly reviewed our manuscript and corrected all instances where "D. melanogaster" was not correctly italicized.

  1. In the abstract, the term "overeating" appears to be overly generalized since the authors are specifically examining the effects of excessive fructose and palmitic acid consumption, rather than the overconsumption of any other diet. Therefore, a more appropriate word should be used.

Response: Indeed, the terms 'overeating' and 'overfeeding' are frequently used in this study. To address this suggestion, we propose replacing these terms with 'overnutrition'. This is because the diet in our experiment included an additional 1% of palmitic acid and 5% of fructose compared to the control group, which provides a more accurate description of the diet administered in this work.

  1. In line 99, please include the references Ganguly, Dey et al., 2021 and May et al., 2019.

Response: We are grateful for the reviewer's valuable input. Following their suggestion, we have supplemented our theoretical framework with two additional references (27 and 28) at line 99.

  1. It would be advantageous to present the upregulated and downregulated genes in a table or as a heatmap. This would enhance comprehension and facilitate data interpretation.

Response: In response to the insightful observation from the reviewer and given the importance of their notes, we have prepared a table cataloguing all genes discussed in this study, distinguishing between those upregulated and downregulated. This table will be included as supplementary information and is referenced at line 117.

Round 2

Reviewer 2 Report

The authors have improved the manuscript by adding some clarification in the text and provided some answers to my questions. There are still, however, a couple of points that might need revisiting but that are not critical. 

Minor points:

-       For future reference, sex differentiation can also be conducted in larvae by visual inspection, without any costly or difficult technique. The authors might want to check the presence of gonads in males around the midsection of the abdominal region. 

-       There are still a couple of asseverations I am not comfortable with, for example, line 215 reads “regarding the reduction of threonine synthesis”. Synthesis was not measured. 

Author Response

Minor points:

-       For future reference, sex differentiation can also be conducted in larvae by visual inspection, without any costly or difficult technique. The authors might want to check the presence of gonads in males around the midsection of the abdominal region. 

Answer: We value the reviewer's feedback and will consider the inclusion of sexing in our future projects.

- There are still a couple of asseverations I am not comfortable with, for example, line 215 reads “regarding the reduction of threonine synthesis”. Synthesis was not measured.

Answer: Due to the helpful reviewer's input, we have modified the phrasing of row 215 to eliminate any potential confusion regarding the results obtained in our study.

Reviewer 3 Report

The current article studies gene expression in Drosophila larvae following feeding a food supplemented with fructose and palmitic acid. However, before undertaking any transcriptome-based analysis it is essential that the authors first validate the dietary condition being used. The authors absolutely needed to characterize whether the larvae are overfeeding. The authors sought to bypass this by using the term 'overnutrition' which does not make sense unless the authors validate it using assays to demonstrate higher nutrient intake. If the larvae decrease their consumption of the high-calorie food, it will not increase their nutritional and caloric intake. The authors absolutely need to support their experimental condition by demonstrating phenotypes of the larvae (such as body weight, size, metabolomics) that support their assumption of the larvae being 'overnutrition'ed.

Author Response

The current article studies gene expression in Drosophila larvae following feeding a food supplemented with fructose and palmitic acid. However, before undertaking any transcriptome-based analysis it is essential that the authors first validate the dietary condition being used. The authors absolutely needed to characterize whether the larvae are overfeeding. The authors sought to bypass this by using the term 'overnutrition' which does not make sense unless the authors validate it using assays to demonstrate higher nutrient intake. If the larvae decrease their consumption of the high-calorie food, it will not increase their nutritional and caloric intake. The authors absolutely need to support their experimental condition by demonstrating phenotypes of the larvae (such as body weight, size, metabolomics) that support their assumption of the larvae being 'overnutrition'ed.

Response: We appreciate your valuable comments and suggestions to improve our manuscript. We understand your concerns and concur that validating the dietary condition is essential for transcriptome-based analysis.

In our study, we have employed the term 'overnutrition' to refer to a high-calorie diet enriched with fructose and palmitic acid, rather than suggesting overfeeding. However, we recognize this may have led to confusion, and we are prepared to revise and clarify this point. Consequently, we have amended the body of the manuscript to explicitly refer to the diet as being "supplemented with palmitic acid and fructose."

We also agree with your suggestion to provide evidence of larval phenotypes that support our 'overnutrition' hypothesis. We conducted the relevant assays and found no significant difference in the weight of larvae fed the normal diet versus those fed the supplemented diet (both were 1.7 mg). This is not unexpected as, in Drosophila, body weight and size are not precise indicators of nutritional status. Therefore, the utilization of alternative markers to assess phenotypic changes has been proposed (DOI: 10.1242/dmm.007948).

Interestingly, we did observe differences in negative geotaxis and survival rates among organisms fed the diet supplemented with palmitic acid and fructose. These findings were presented at the 2022 Genetics Society Congress (abstract 972B). Further details can be found in the conference proceedings, available at the following link: https://genetics-gsa.org/drosophila-2022/program-and-abstract-books/. Based on this evidence, we affirm that the supplemented diet we used does have an impact on phenotypic data, consistent with the transcriptomic changes we observed.

Once again, we thank you for your constructive feedback and trust that these clarifications and amendments will enhance the quality of our manuscript.

Round 3

Reviewer 3 Report

I am satisfied with how the authors addressed the concerns I had raised.